# Factors Influencing the Adoption of E-Government Services: A Study among University Students

**Carlos Alberto Méndez-Rivera** [1], **Orfa Nidia Patiño-Toro** [1], **Alejandro Valencia-Arias** [2,*] 
**and Diana María Arango-Botero** [1]

1. Facultad de Ciencias Económicas y Administrativas, Instituto Tecnológico Metropolitano, Medellín 050005, Colombia; carlosmendez3537@correo.itm.edu.co (C.A.M.-R.); orfapatino@itm.edu.co (O.N.P.-T.); dianaarangob@itm.edu.co (D.M.A.-B.)
2. Escuela de Ingeniería Industrial, Universidad Señor de Sipán, Chiclayo 14001, Peru
* Correspondence: valenciajho@crece.uss.edu.pe; Tel.: +57-3002567977

**Abstract:** The digitization of government services meets the expectations of the younger population, who are accustomed to widespread ICT use. It offers transparency, speed, efficiency, and trust, supported by international organizations. This research aims to identify the factors that influence the adoption of e-government services among university students affiliated with the District Mayor's Office of Science, Technology, and Innovation in Medellín. The study involved surveying a sample of 403 students to examine their intention to adopt e-government services using structural equation modeling. The results highlight the significant impact of perceived usefulness on attitude towards use and, subsequently, on the intention to use e-government. Notably, subjective norm has the least influence on intent to use. The study also underscores the potential of mobile e-government as a promising option, considering the widespread access to smartphones in emerging economies. Cultural factors, usability, data privacy, lack of trust in governments, and entrenched mentalities emerge as barriers to e-government implementation. In conclusion, the findings shed light on the challenges posed by inadequate infrastructure, digital literacy gaps, resistance to change, and cultural factors that impede e-government adoption. Non-adoption would result in technological setbacks, negative indicators, and inefficiencies. Moreover, recognizing the pivotal role of university students in promoting e-government tools among their peers and relatives, this study emphasizes the importance of their involvement.

**Keywords:** e-government; student; university; factors; adoption; innovation

## 1. Introduction

ICTs are considered pioneering tools with which to promote the better execution of government programs and services which, in turn, empower citizens through greater access to information, the application of more efficient government management processes, greater transparency and accountability, and the mitigation of corruption risks (Neupane et al. 2015). Furthermore, according to Xu (2010), the innovative provision of e-services will potentially be the heart of innovation in the public sector. For the author, opportunities are currently being wasted due to the lack of a clear vision and generic reference models for the provision of e-services.

The modern government is in dire need of better functional systems with which to introduce advanced citizen-centric features quickly, safely, and successfully. Governments have been experimenting and have found innovative ways to offer citizen-centred services (Manocha et al. 2018). For Jayashree and Marthandan (2010, p. 2205), "one of the most important aspects of e-government is to transform the physical society into e-society where the citizens will have seamless interactions with the Government and its functions and services".

In this context, the adoption of e-government tools could eventually reduce administrative costs and the time spent by public servants to perform repetitive tasks, thus offering greater transparency to the public administration, improving the current performance of services and public sector procedures, and significantly expanding access to those services (Muñoz 2014).

In this way, e-government initiatives are becoming common throughout the world, but there is still a lack of conceptual elements to understand their development, implementation, and consequences (Córdoba-Pachón and Orr 2009). Notably, e-government is dedicated not only to providing services to citizens but also to improving the efficiency, transparency, and accountability of the public sector in government functions and reducing the costs of public administration. "In fact, the ultimate goal of E-government is to be able to offer public services to citizens in an efficient and cost-effective manner, which is also the good governance maxim" (Madzova et al. 2013, p. 158).

In recent years, society has demonstrated a growing need to carry out tasks quickly, efficiently, and economically. This has led to the development of numerous devices, tools, and procedures that have become accessible for the betterment of humankind. Information and Communication Technologies (ICT) play a crucial role in facilitating access to these tools, eliminating costs associated with travel, time, and administrative procedures (Jha and Sarangi 2022).

The COVID-19 pandemic declared by the WHO in March 2020 has consolidated using virtuality as the main way to carry out daily activities such as teleworking, telehealth, tele-education, online shopping, and online procedures. This forced public and private entities to improve their ICT capacity and infrastructure to cope with the exponential increase in virtual interactions due to the restrictions of the then-existing health emergency (Hai et al. 2021).

E-government stands out as a key tool in the new millennium, enabling the provision of government services independently in terms of time, distance, and organizational complexity. The adoption of e-government is a goal in the broadband policies for Latin America and the Caribbean of the OECD, which emphasize the need to improve the provision and quality of digital government, strengthen the management of government information services, and leverage technology and innovation to improve city organization (OECD/IDB 2016).

However, the implementation, adoption, and use of e-government have represented a challenge, especially in developing countries, due to limitations in infrastructure, demographic difficulties, low digital literacy, and cultural influence (Haughton and Barnes 2023). In Colombia, it is crucial to understand the factors and variables involved in the digital government tools' adoption so that the central, departmental, and municipal governments can make decisions focused on promoting their adoption and universalization (Ramirez-Madrid et al. 2022).

At the local level, the universalization of Internet service in Medellín is progressing slowly, especially in low-income sectors, which hinders the possibility of achieving a fully electronic government in the short term. The city's youth, particularly university students, play an essential role in the adoption and use of e-government (Portela and Rossenver 2022). Therefore, it is relevant to investigate the motivations that drive these young people to access e-government services, as they are likely to use these tools in their professional lives more frequently. These motivations can be understood as related factors and variables. Through a modification of the technology acceptance model (TAM), a model is proposed that seeks to identify the most relevant factors, as well as their correlation and relationship with some notable variables in this context.

Digital government by design is a form of governance that uses digital technologies to rethink and redesign public processes. The goal is to simplify procedures and create new channels of communication and participation for citizens (Salazar Espinoza 2022). Data-driven government, on the other hand, implements a wide range of tools, standards, and services that enable teams to focus on user needs to design and deliver public services more

efficiently and effectively (Gomis-Balestreri 2017). A user-directed government prioritizes people's needs and convenience when shaping processes, services, and policies and adopts inclusive mechanisms that allow this to happen (OECD/IDB 2016). In Colombia, migration to digital government is also measured with the digital government indices, which are based on the analysis of data collected through the Single Form of Management Progress Reports (FURAG).

According to Sánchez and Corral (2014), digital tools are "all those intelligible software or programs found on computers or devices" (p. 2) that allow multiple tasks and activities to be carried out through interactions with technology that make communicating more efficient. For Daily and Peterson (2017), owing to digital tools, people interact more seamlessly, and data are shared openly. Thus, the adoption of technological tools implies the interaction of an individual with other people or electronic devices through software, which allows them to carry out various activities and tasks in areas such as state, student, work, business, among others. Intuitively, these tools make it easy to perform various activities and tasks in everyday life (Ali et al. 2020).

In this sense, the adoption of innovations is a highly relevant topic in the literature that seeks to explain consumer-user behavior. Innovation adoption models are highly relevant because clarity regarding the factors that determine the individual decision to adopt or not adopt an innovation can reveal the process by which an individual begins to consume a good, a service or an idea (López-Bonilla and López-Bonilla 2010).

Although there is recognition of the relevance of electronic government and its contribution to improving the accessibility of information to individuals, the transparency of processes, and the effectiveness in the provision of government services, among other contributions, the existence of a gap in knowledge concerning the motivations or specific reasons responsible for promoting the use of these resources in young university students is evident (Borja Acosta 2022). Thus, the understanding of these factors is essential for the broad and effective promotion of the adoption of these technological tools in the context of public entities, as well for their efficacy in helping government authorities make informed decisions, the official order for their materialization, the massification of their use, and the exploitation of the benefits of e-government by citizens (Glyptis et al. 2020).

Likewise, there is limited information on how the adoption by individuals of digital tools made available by government agencies has changed the interaction between the people and these entities, generating trust and increasing user participation in the services of official institutions. In this sense, and considering the available empirical information, the objective of this study is to identify the factors that influence the adoption of digital government services among young university students belonging to the universities affiliated to the Mayor's Office of Medellin (Alcaldía de Medellín 2020).

Based on this study and the identification of these elements, we can contribute to improving the assimilation of the phenomenon and offer important information in favor of the execution and dissemination by government organizations of the e-government services they have, in addition to the possible expansion of knowledge on the subject among university students, taking into account that an substantial portion of the young citizens of Medellín are university students, i.e., potential direct actors in the construction of the country and regular Internet users (Alcaldía de Medellín 2020).

## 2. Literature Review

### 2.1. Conceptual Framework

For the Secretariat for Political Affairs of the OAS, e-government involves "the use of ICT Information and Communication Technologies by government institutions to qualitatively improve the services and information offered to citizens, increase the efficiency and effectiveness of public management and substantially increase the transparency of the public sector and citizen participation" (OEA 2021). For the UN, e-government involves "the use of Information and Communication Technologies to provide independent government services in time, distance and organizational complexity" (Naser and Concha 2011) To

summarize, e-government can be understood as the use of the internet to remotely access the services and operations offered by government entities.

### 2.1.1. E-Government

According to Naser and Concha (2011), the importance of e-government lies in the need to streamline, optimize, make flexible and transparent, and lower the cost of processes and activities of the public system. This implies developing models adapted to the needs of the government by creating platforms that solve problems of interoperability, compatibility, access, and security. The objective is to improve the information services offered to citizens and organizations, simplify institutional support processes, and facilitate channels that promote transparency, citizen participation, online services, and distance training in an open and equitable manner (Naser and Concha 2011).

### 2.1.2. E-Government in Colombia

According to the provisions of the Digital Government Manual of the Republic of Colombia—MinTIC (2018), e-government is characterized by the use of information and communication technologies (ICTs) to access e-government services, as well as the provision of services by governments in an agile and efficient way and the participation of citizens in the decision-making process within a framework of transparency, all of which directly facilitate deliberative democracy.

### 2.1.3. Adoption of Innovation

Rogers (1962) defines the adoption of innovation or technology as the process of accepting, integrating, and utilizing new technologies. This social phenomenon involves communicating innovation through specific channels to the intended members of the social system. As per Rogers' study, there are five types of users based on their adoption timing:

The first group comprises innovators, i.e., the initial users, those who embrace innovation. They are characterized by their willingness to take higher risks and tend to be younger, to hold prominent social status, and to be financially supported even in case of failure.

Visionaries constitute the second category, demonstrating a high degree of leadership and a prestigious social status. Unlike Innovators, they adopt innovations more discreetly.

The third group, the early majority, adopts innovations after a certain period of observation. They proceed more cautiously in the adoption process and maintain an above-average social standing. Individuals in the early majority often have connections with early adopters and occasionally display leadership qualities.

The late majority represents the fourth category of users, i.e., those who adopt innovations much later than the average members of society. They exhibit significant scepticism towards adopting new technologies, typically possess a social status below the average, and face financial uncertainty.

Lastly, the sceptics or laggards constitute the final group of users, i.e., those who are resistant to change and are often of advanced age. They lack thought leadership and hold an aversion to agents of change.

In addition to Rogers' classification, various models of innovation adoption provide valuable insights and explanations regarding how people embrace new technologies and innovations within diverse social and cultural contexts. Each of these models possesses distinct strengths and practical applications, contributing to the comprehensive study of technology adoption in different domains.

One such model is the theory of reasoned action (TRA), introduced by Ajzen and Fishbein in 1980. TRA suggests that "each individual is aware of their behaviors; therefore, when they carry out a behavior, they do so sensibly" (Ajzen and Fishbein 1980). This theory has been employed to investigate the acceptance of Islamic financial technology (FinTech) banking services in Malaysia, as evidenced by a study conducted by Shaikh et al. in 2020 (Shaikh et al. 2020).

Another significant model is the unified theory of acceptance and use of technologies (UTAUT). This model explains the intention to use an information system and subsequent behavior by considering four main constructs: expectations of performance, expectations of effort, social influence, and facilitating conditions (Venkatesh et al. 2003). Recently, UTAUT was applied to study the sustainable adoption of e-government services in Northern Iraq (Zeebaree et al. 2022).

Social cognitive theory (SCT), formulated by Wood and Bandura in 1989, is another influential model based on four principles: observational learning, the possibility of learning without immediate behavioral change, the role of consequences in learning, and the significance of learning and cognition (Wood and Bandura 1989). SCT has been employed to investigate factors influencing citizens' decisions to participate in the Internet (Khoirunnida et al. 2017).

Derived from TRA, the theory of planned behavior (TPB), proposed by Ajzen in 1991, focuses on predicting non-conscious or voluntary behaviors that individuals do not entirely control (Ajzen 1991). This theory has been utilized to explore the adoption of e-government systems by non-adopters in the Indian context (Rana et al. 2016).

The widely accepted technological acceptance model (TAM), introduced by Davis in 1986 and later extended by Venkatesh and Davis in 1996, remains a fundamental model in technology adoption research (Davis 1986; Venkatesh and Davis 1996). It enables the prediction of technology acceptance and usage by analyzing three essential factors: perceived usefulness (PU), perceived ease-of-use (PEOU), and perceived enjoyment (PE). Researchers have applied TAM in a literature review pertaining to its applicability in the Peruvian agricultural export sector with respect to e-government, uncovering 10 models and 36 factors associated with perceived usefulness and ease-of-use (Salas et al. 2022).

TAM2 (technological acceptance model 2), an extension of the original TAM by Venkatesh and Davis in 2000, incorporates three additional constructs: subjective norms, voluntariness of use, and image (Venkatesh and Davis 2000). This expanded model enhances the understanding of technology adoption by considering social aspects and individual perceptions related to a common system.

A hybrid model, the C–TAM–TPB Model (Yayla and Hu 2007), combines elements of TAM and TPB while integrating normative and control aspects of perceived behavior. This model offers improved predictive capabilities for determining the intent to use a system.

TAM 3 (Venkatesh and Bala 2008) further extends TAM 2 by incorporating the explanatory factor of perceived ease-of-use through the linkage factor, considering individual personality traits. Additionally, it introduces adjustment factors to better reflect the progress of the adoption process and includes the use of the system as a moderating variable.

The decomposed theory of planned behavior (DTPB), introduced by Taylor and Todd in 1995, is a combination of the IDT and TPB, with elements from TAM such as perceived usefulness and perceived ease-of-use (Taylor and Todd 1995). DTPB offers enhanced explanatory power regarding behavioral intention and a deeper understanding of the relationships between components and their antecedents. This theory has recently been applied in the South Asian region to determine the intention to use e-government services (Zahid and Din 2019).

### 2.2. Development of Hypotheses

Technology adoption models (TAMs) are based on descriptions of the information processes that an individual executes to lead them to accept or reject the innovation they are considering through a subjective evaluation of perceptions of an object and the perspectives generated through the use of that object. These perspectives are supported by beliefs of behavior, knowledge, and sometimes, affect, which are widely described in the existing innovation adoption models, where the generalization of the model is highlighted and whose foundation originates in the theory of reasoned action (López-Bonilla and López-Bonilla 2010). The most relevant variables are defined below, and they served as input for the design of the survey.

### 2.2.1. Attitude towards Use (AHU)

According to the decomposed theory of planned behavior developed by Taylor and Todd (1995), the attitude towards the use of new technologies is conditioned by the relative advantage, complexity, and compatibility of the user with that technology. In addition, Wu and Chen (2005) indicate that attitude towards use is a significant predictor of behavior and is a relevant factor that motivates the decision to continue using electronic channels.

### 2.2.2. Perceived Ease-of-Use (FUP)

According to the technological acceptance model (TAM) developed by Venkatesh and Davis (1996), perceived ease-of-use can be defined as the perception a user has of expending less effort when using a particular system when seeking a result.

### 2.2.3. Intention of Use (IU)

According to the C–TAM–TPB Model developed by Yayla and Hu (2007), prediction is achieved through the analysis of normative or social aspects and the control of perceived behavior. The model seeks to measure, with the greatest possible reliability, the intention that individuals have to use a technology or a system. According to Rojas (2022), the intent to use is strongly linked to social influence—that is, the power that people important to the individual have and which influences their decision to use the system.

### 2.2.4. Subjective Norm (NS)

According to the decomposed theory of planned behavior (DTPB) developed by Taylor and Todd (1995), a subjective norm involves normative reasoning in which the pressure of society is perceived by the person who is going to execute a behavior towards a technology or system. This theory also considers the motivation of the individual to change that behavior depending on the collective thought or of other people.

### 2.2.5. Actual Use of the System (UR)

This variable is a measure of the actual use of a technology or a system over time. It is linked to future uses of that technology or system and, therefore, to a successful adoption process Davis (1986).

### 2.2.6. Perceived Usefulness (UP)

According to the technological acceptance model (TAM) developed by Davis (1986), this variable is the perceived level that a user has when they believe that they or their performance will stand out through the use of a particular technology or system. Based on this information, the following hypothesis is proposed:

**H1:** *Perceived utility (UP) has a positive and significant effect on attitude toward use (AHU) with respect to e-government.*

For Davis (1986), in their technological acceptance model (TAM), perceived utility (UP) is the perceived level that a user has when he or she believes that using a particular technology or system will distinguish him or her or improve his or her performance. If the perceived usefulness is significant and positive for the individual, it is highly likely that this will positively influence their attitude towards use, as confirmed by recent studies by Reyes and Castañeda (2020). In other words, an increase in the perceived utility by the user will generate a positive effect on the attitude towards the use of the system or technology (Ballinas et al. 2013). Therefore, the following hypothesis is proposed:

**H2:** *Perceived ease-of-use (FUP) has a positive and significant effect on attitude toward use (AHU) with respect to e-government.*

Similarly, Davis (1986) define perceived ease of use (PEU) as a user's perception of less effort required when using a particular system in search of a result. If the individual perceives that the system is easier to use, there is a great possibility that his attitude towards

its use will be greater, as confirmed by the studies of Muñoz (2008) and Bregashtian and Herdinata (2021). Put another way, the higher the perception of the system's ease-of-use, the more positive the attitude towards its use will be. It is important to note that in the TAM (technology acceptance model), the attitude towards system use is based on the perceived usefulness and perceived ease-of-use variables. Following Muñoz (2014), perceived ease-of-use is influenced by effectiveness and instrumentality. Therefore, this ease-of-use effect is directly related to the attitude. Accordingly, the following hypothesis is proposed:

**H3:** *Perceived utility (UP) has a positive and significant effect on intention to use (IU) with respect to e-government.*

Perceived usefulness is based on studies of motivations and expectations. This is why individuals use systems and technologies only if they perceive that their use will help them achieve their objectives (Taylor and Todd 1995). According to Davis (1989), perceived utility directly influences use through intention to use. In the same sense, several studies support the direct relationship between perceived utility and intention to use in TAM models, such as those of Ferrer and Bravo (2012), Matute et al. (2015), and Tavera and Londoño (2014).

In the same sense, for Taylor and Todd (1995), the attitude towards the use of new technologies is conditioned by the relative advantage, complexity, and compatibility of the user with this technology. Furthermore, Wu and Chen (2005) state that attitude toward use becomes a significant predictor of behavior and is a relevant factor that motivates the intention to continue using electronic channels. Salas-Rubio et al. (2021) reveal that the influence of the attitude towards system use is related to the intention to use, and this has been demonstrated from various perspectives, such as psychological, technological, and managerial, among others. Therefore, the study proposes the following hypothesis:

**H4:** *Attitude toward use (AHU) has a positive and significant effect on intention to use (IU) with respect to e-government.*

According to Davis (1986), it is argued that the attitude towards using a system is determined by the perceived utility and the perceived ease-of-use. The easier it is to interact with the technology or system, the higher the perceived effectiveness for the user should be, and therefore, the greater the willingness to use it, understood as a higher intention to use it (Urquidi Martin and Calabor Prieto 2014). From a study by Hajli (2012) on electronic commerce, the concept that the influence of ease-of-use on the usefulness of a system is real and meaningful is extracted, which leads to the following hypothesis:

**H5:** *The perceived ease-of-use (FUP) has a positive and significant effect on the perceived utility (UP) with respect to e-government.*

On the other hand, actual use of the system (UR) is a variable that seeks to measure the actual use of a technology or system over time, which is also related to the future use of that technology or system and, therefore, to a successful adoption process (Davis 1986). According to this statement, it can be inferred that the successful adoption process is achieved through manifest intention to use; if the user has a greater willingness to use the system, it will imply a higher usage of the system. Therefore, the following hypothesis is proposed:

**H6:** *The intent to use (IU) has a positive and significant effect on the actual use of the system (UR) with respect to e-government.*

According to the decomposed theory of planned behavior (DTPB) by Taylor and Todd (1995), the subjective norm (NS) is a form of normative reasoning where the pressure of society is perceived by the person who will perform the behavior towards the technology or system. Added to this is the motivation of the individual to change that behavior or not depending on collective thinking or that of other people. In addition to this, the studies by Venkatesh and Davis (2000) allowed for a direct relationship between subjective norm and intention to use, based on four studies on technological acceptance in face-to-face environments. Based on the above, the following hypothesis is proposed:

**H7:** *Subjective norm (NS) has a positive and significant effect on intention to use (IU) with respect to e-government.*

According to Rojas (2022), the intention to use is strongly related to the social influence—that is, the power that important people have over the individual which influences his decision to use the system. Particularly, the existence of a positive relationship between the intention to use e-government and the actual usage of this system indicates that individuals with a higher intention to use electronic government are more likely to actively participate and use the resource in practice.

Based on the generated hypotheses and the variables identified from the selected literature, the following initial model is proposed in Figure 1.

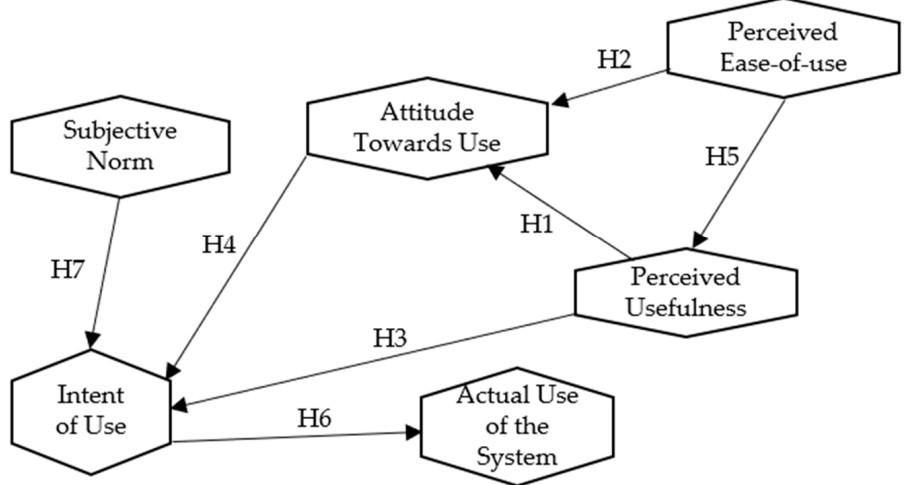

**Figure 1.** Proposed model. Note: own elaboration.

### 3. Materials and Methods

This study presents a quantitative approach with a correlational scope to understand the degree of association among the analyzed variables in a specific context (Hernández et al. 2010). In this case, the analysis was conducted around the variables related to the adoption of e-government in students from three higher education institutions (HEIs) affiliated with the District Mayor's Office of Science, Technology, and Innovation of Medellín during the academic periods 2022–1 and 2022–2 (Colegio Mayor, ITM and Pascual Bravo).

#### 3.1. Population and Sample Calculation

The sample for the quantitative analysis was determined using the known size of the potential study population: 37,855 enrolled students in the three universities (El Colombiano 2022). As the total size of the population was known, the formula for a finite population was used. Taking into account a margin of error of 5% and a confidence level of 95%, the sample size calculated using Equation (1) is 381 students. Table 1 shows the sociodemographic information of the sample, which was composed by 403 university students.

Equation (1):

Formula for calculating the number of sample elements (finite population).

$$n\text{opt} = \frac{N * Z2 * p * q}{d2 * (N - 1) + Z2 * p * q} \tag{1}$$

Source: Marín (2017).

**Table 1.** Sociodemographic information of the sample.

| Variable | Level | Frequency (n = 403) | Percentage |
|---|---|---|---|
| University Institution | Colegio Mayor | 76 | 18.86% |
| | ITM | 251 | 62.28% |
| | Pascual Bravo | 76 | 18.86% |
| Current semester | 1 | 23 | 5.71% |
| | 2 | 126 | 31.27% |
| | 3 | 40 | 9.93% |
| | 4 | 72 | 17.87% |
| | 5 | 41 | 10.17% |
| | 6 | 25 | 6.2% |
| | 7 | 12 | 2.98% |
| | 8 | 19 | 4.71% |
| | 9 | 15 | 3.72% |
| | 10 | 16 | 3.97% |
| | SN/NR | | 3.47% |
| Gender | Female | 223 | 55.33% |
| | Male | 163 | 40.45% |
| | SN/NR | 17 | 4.22% |
| Occupation | Employee | 240 | 59.55% |
| | Independent | 57 | 14.14% |
| | Unemployed | 98 | 24.32% |
| | SN/NR | 8 | 1.99% |
| Age range | Between 16–27 years | 251 | 62.28% |
| | Between 28–40 years | 122 | 30.27% |
| | Between 41–57 years | 11 | 2.72% |
| | SN/NR | 19 | 4.71% |
| Socioeconomic stratum | 1 | 49 | 12.16% |
| | 2 | 160 | 39.70% |
| | 3 | 143 | 35.48% |
| | 4 | 32 | 7.94% |
| | 5 | 6 | 1.49% |
| | SN/NR | 13 | 3.23% |

*3.2. Instrument*

A survey was chosen as the data collection method, using a self-administered questionnaire. This instrument consisted of initial questions for characterizing the surveyed population (institution, program, gender, socioeconomic status, age, occupation, among others). Additionally, five multiple-choice questions and twelve statements in a Likert scale were included to assess the level of agreement or disagreement or rate the perceived importance of respondents' beliefs or perceptions regarding the attributes of the topic (Chu and Choi 2000; Heo et al. 2022; Likert 1932). The scale used corresponds to the following possible responses: 5: strongly agree; 4: agree; 3: neither agree nor disagree; 2: disagree; and 1: strongly disagree. Furthermore, the questionnaire includes two open-ended questions associated with the advantages and disadvantages of using e-government and possible elements to consider in its implementation.

For data analysis, SPSS version 22 was used, which is a comprehensive statistical analysis software designed by IBM. This software ensured the required fluidity and precision in the obtained data analysis. The adjustment of correlations in the initial model were performed using structural equation modeling, considered one of the most relevant tools for studying causal relationships in non-experimental data (Medrano and Muñoz-Navarro 2017).

### 3.3. Procedure

Before the questionnaire administration, a pilot test was conducted with 15 participants to establish the clarity of the questionnaire design, avoid potential limitations or biases, and determine the relevance of the variables used as the basis for data analysis in the context of e-government adoption (Castrillón and Mandakovic 2010). The designed and approved instrument from the pilot test was administered in the presence of one of the authors of this study to address any concerns that participants may have had. Emphasis was placed on the importance of responding with autonomy and tranquility, as the process was anonymous (Ortiz et al. 2014). Additionally, to encourage honest and unbiased responses from the participants, simple and non-redundant questions were used and mixed throughout the instrument to avoid systematic responses.

One possible limitation in the data collection process was selection bias because some groups of students had a lower probability of being selected. As the surveys were conducted between 6:00 p.m. and 10:00 p.m, many daytime students do not take courses during the mentioned hours, thus remaining ineligible for the sample. Another limitation was the probability of non-response with respect to certain items. To minimize this, each construct had several similar and related points in case it became necessary to eliminate any. Finally, it is important to take into account that the questionnaire was administered only to students from public institutions, which limits the generalizability of the results. This is considering that private institutions may have better technological infrastructure for student education, which could lead to differences in technology adoption.

## 4. Results

The findings of this study are based on a factor analysis, aiming to reduce the dimensions of multivariate random variables in order to identify factors. This is used as part of the validation process of scales or measurement instruments, serving as a tool by which to analyse the internal structure of the data as sources of construct validity (Pérez 2020). In other words, this technique serves to measure the relative association (correlation) between variables (Kipfer 2021).

### 4.1. Common Method Bias

To verify the absence of common method bias, Harman's one-factor test was conducted. By reducing the dimensionality to a single factor, an explained variance of 30.53% was obtained, which is below the cutoff point of 50% (Podsakoff and Organ 1986), commonly accepted in such verifications. Therefore, it can be concluded that common method bias is not a problem in our research.

### 4.2. Construct Validity

To verify that the items were part of an underlying factor structure, the KMO coefficient was used, as well as the Bartlett's test of sphericity. KMO is defined as the measure of the Kaiser–Meyer–Olkin sampling adequacy and determines whether the partial correlations between the variables are small. In this test, values ranging from 0.05 to 1 should be considered as the basis for analysis. Similarly, Bartlett's test of sphericity determines whether the correlation matrix is an identity matrix, i.e., whether the factorial model is inadequate. The resulting correlation matrix is presented in Table 2, which shows that all the KMO values are between 0.500 and 0.870, meeting the level of sample fit, and the Bartlett's test results were all significant.

To confirm the internal consistency of the items, Cronbach's alpha coefficients were calculated, which describe how close or how few related variables are in the data (Godoy 2022). Regarding the meaning of the alpha values, Renova et al. (2021) state that alpha values greater than or equal to 0.90 are excellent, between 0.80 and 0.90 are good, between 0.70 and 0.80 are acceptable, between 0.60 and 0.70 are questionable, between 0.50 and 0.60 are poor, and less than 0.50 are unacceptable. For this study, as all alpha index values were above 0.8 (Table 3), confirming the condition of stability and internal consistency.

**Table 2.** Correlation matrix: KMO index and Bartlett's test of sphericity.

| Factor | KMO | Bartlett |
|---|---|---|
| Perceived Ease-of-Use | 0.809 | 0.000 |
| Intent to Use | 0.751 | 0.000 |
| Subjective Norm | 0.500 | 0.000 |
| Attitude Towards Use | 0.870 | 0.000 |
| Perceived Usefulness | 0.720 | 0.000 |
| Actual Use of the System | 0.741 | 0.000 |

**Table 3.** Factor loadings and Cronbach's alpha for the theoretical model.

| Factor | Item | Factorial Load | Average Factorial Load | Cronbach's Alpha |
|---|---|---|---|---|
| Perceived Ease-of-Use | FUP 1 | 0.813 | 0.767 | 0.893 |
| | FUP 2 | 0.794 | | |
| | FUP 3 | 0.753 | | |
| | FUP 4 | 0.804 | | |
| | FUP 5 | 0.733 | | |
| | FUP 6 | 0.707 | | |
| Intent to Use | IU 1 | 0.852 | 0.810 | 0.886 |
| | IU 2 | 0.802 | | |
| | IU 3 | 0.772 | | |
| | IU 4 | 0.813 | | |
| Subjective Norm | NS 1 | 0.929 | 0.929 | 0.943 |
| | NS 2 | 0.929 | | |
| Attitude Towards Use | AHU 1 | 0.826 | 0.842 | 0.927 |
| | AHU 2 | 0.908 | | |
| | AHU 3 | 0.909 | | |
| | AHU 4 | 0.866 | | |
| | AHU 5 | 0.701 | | |
| Perceived Usefulness | UP 1 | 0.880 | 0.895 | 0.930 |
| | UP 2 | 0.924 | | |
| | UP 3 | 0.880 | | |
| Actual Use of the System | UR 1 | 0.778 | 0.787 | 0.869 |
| | UR 2 | 0.793 | | |
| | UR 3 | 0.852 | | |
| | UR 4 | 0.725 | | |

Once the factorial structure and internal consistency were confirmed, convergent and discriminant validity was evaluated. Convergent validity is a subtype of construct validity, a test designed to measure a particular construct, seeking to prove that the relationship between constructs exists without doubt. Convergent validity is confirmed if the correlation coefficient is equal to or greater than 0.50 (Carlson and Herdman 2012). This suggests that the initial factors necessary to represent the original data should be extracted. Table 3 shows the factor loadings to establish the correlation between each item and each factor analyzed.

According to the discriminant validity, which is used to assess whether two variables that are supposed to be unrelated are, in fact, unrelated, the confidence intervals for discriminant validity should never include 1 (Carlson and Herdman 2012). Table 4 shows the correlation matrix with the intervals obtained in SPSS. As can be observed, there is no value of 1, confirming that the model is valid.

**Table 4.** Confidence intervals for discriminant validity.

| Factor | FUP | IU | NS | AHU | Up | UR |
|---|---|---|---|---|---|---|
| Perceived Ease-of-Use | - | | | | | |
| Intent to Use | [0.526; 0.667] | - | | | | |
| Subjective Norm | [0.314; 0.489] | [0.472; 0.631] | - | | | |
| Attitude Towards Use | [0.541; 0.681] | [0.555; 0.694] | [0.405; 0.573] | - | | |
| Perceived Usefulness | [0.560; 0.693] | [0.503; 0.661] | [0.377; 0.551] | [0.603; 0.741] | - | |
| Actual Use of the System | [0.440; 0.605] | [0.536; 0.685] | [0.349; 0.523] | [0.520; 0.670] | [0.495; 0.652] | - |

### 4.3. Fitting of a Structural Equation Model and Hypothesis Testing

The methodology of structural equation modelling (SEM) proposes a structural model with which to verify the hypothesized relationships. One of the advantages of SEM is the inclusion of an error term in the measurement of the latent variables, which is not included in traditional regression models where error-free measurements are assumed. The SEM assumptions include multivariate normality, no systematic missing data, adequate sample size, and correct model specification.

To verify the assumption of normality in the data, kurtosis and skewness measures were calculated. According to Hu and Bentler (1999), acceptable values to confirm normality are between −2 and 2. The skewness coefficient values for each item or question ranged from −0.2 to 0.2, while the kurtosis coefficient values ranged from 1.8 to 3.0. Therefore, the "Diagonally Weighted Least Squares" (DWLS) was used as an estimator method to adjust the structural equation model with the help of the "sem" function in the lavaan package of the R software. Studies supporting the use of this method for non-normality situations or when adjusting ordinal variables include DiStefano and Morgan (2014) and Mindrila (2010), among others.

Regarding the presence of outliers (values that deviate more than three times the interquartile range from both the lower quartile and the upper quartile), 52 records were identified and subsequently removed, resulting in 351 observations for model adjustment.

To evaluate the fit of the initial structural equation model (Figure 1), several indices were calculated, as suggested by different authors (Hair et al. 2010; Hu and Bentler 1999). According to Hair et al. (2010), CFI and TLI values above 0.9 indicate a good fit. Hu and Bentler (1999) suggest that the quotient of the chi-square statistic over the degrees of freedom should not exceed the value of 3, and for the SRMR and RMSEA indices, values below 0.08 and 0.06 are deemed adequate to indicate a good fit. Table 5 presents the calculations for the initial model.

**Table 5.** Indices to evaluate the goodness of fit of the initial model.

| Initial | Chi Square/Degrees of Freedom | CFI | TLI | SRMR | RMSEA |
|---|---|---|---|---|---|
| Model | 5.039 | 0.813 | 0.788 | 0.078 | 0.107 |

As seen in Table 5, none of the indices meet the cut-off point; therefore, to improve the fit of the model, all the items that reflect each of the latent variables or constructs were reviewed, and the items AHU5, FUP1, FUP2, and IU4 were eliminated (Figure 2). Table 6 presents the goodness-of-fit indices for the modified model.

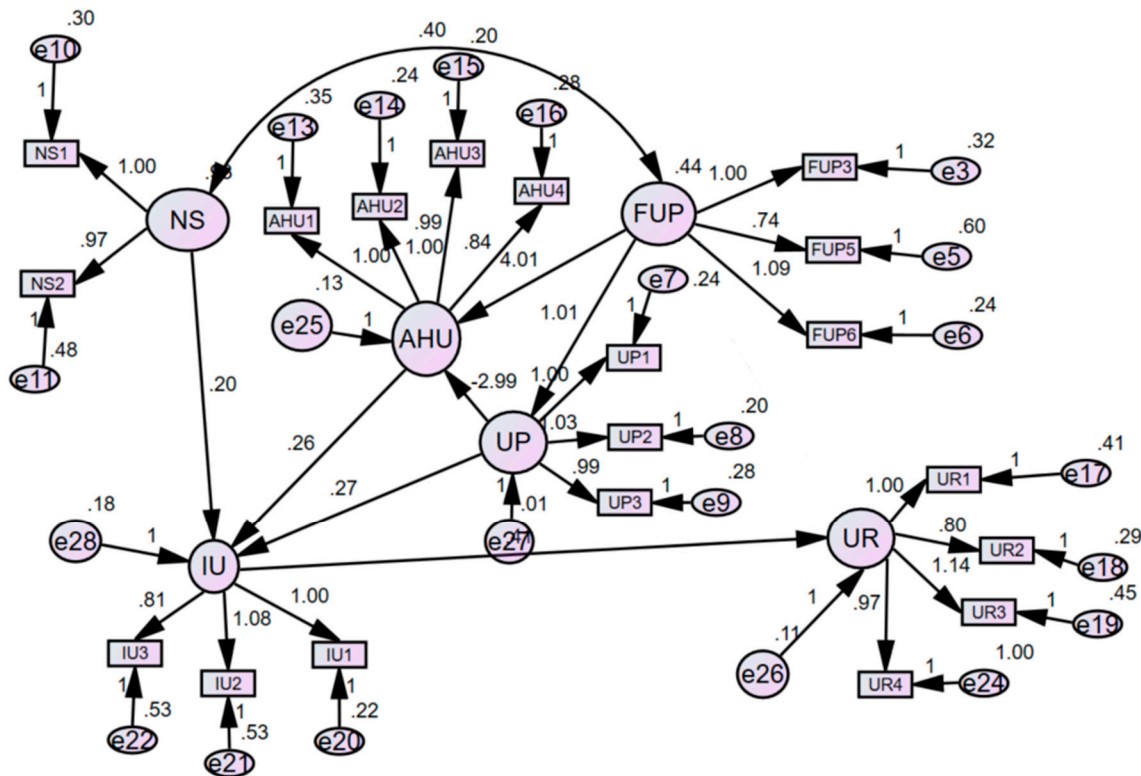

**Figure 2.** Modified model.

**Table 6.** Indices to evaluate the goodness of fit of the modified model.

| Modified | Chi Square/Degrees of Freedom | CFI | TLI | SRMR | RMSEA |
|---|---|---|---|---|---|
| model | 3.61 | 0.904 | 0.905 | 0.056 | 0.086 |

With the modifications made, it was possible to improve the fit of the model—that is, from not meeting any of the five proposed indices to meeting three of them (CFI, TLI, and SRMR). Based on this adjustment, the hypotheses were tested.

From the results in Table 7, it can be concluded that two of the seven hypotheses posed by the model are not supported at a significance level of 5%. These hypotheses correspond to a positive effect of perceived usefulness on attitude towards use and a positive effect of perceived ease-of-use on attitude towards use, with p values of 0.646 and 0.548, respectively. Otherwise, all other hypotheses are supported at a level of significance lower than 0.001, suggesting a high degree of significance. The results, therefore, suggest that the greater the perceived usefulness, attitude towards use, and subjective norm, the greater the intent to use.

**Table 7.** Estimates and hypothesis testing.

| Hypothesis | Estimated Beta | Standard Error | z Value | p Value | Conclusion |
|---|---|---|---|---|---|
| H1. Perceived Usefulness → Attitude Towards Use | −2.99 | 6.514 | −0.460 | 0.646 | not supported |
| H2. Perceived Ease-of-Use → Attitude Towards Use | 4.01 | 6.681 | 0.601 | 0.548 | not supported |

**Table 7.** *Cont.*

| Hypothesis | Estimated Beta | Standard Error | z Value | *p* Value | Conclusion |
|---|---|---|---|---|---|
| H3. Perceived Usefulness → Intent to Use | 0.27 | 0.093 | 2.907 | 0.004 | supported |
| H4. Attitude Towards Use → Intent to Use | 0.26 | 0.071 | 3.731 | 0.000 | supported |
| H5. Perceived Ease-of-Use → Perceived Usefulness | 1.01 | 0.067 | 15.171 | 0.000 | supported |
| H6. Intent to use → Actual Use of the System | 0.41 | 0.084 | 4.852 | 0.000 | supported |
| H7. Subjective rule → Intent to use | 0.2 | 0.048 | 4.070 | 0.000 | supported |

## 5. Discussion

This research, which is characterized by selecting a population with significant electronic preparation, revealed that the first two hypotheses were not supported for the proposed model. This means that neither perceived usefulness nor perceived ease-of-use have a significant positive influence on attitude towards use among the surveyed students. It is highly likely that, being predominantly a young population, it can be assumed that they already have a positive attitude towards the use of e-government, which would explain why their perception of utility/benefit and ease-of-use are not determining factors for such an attitude. These results align with the findings of ElKheshin and Saleeb (2017), who assert that contrary to the original TAM (technology adoption model) proposal, perceived usefulness did not directly affect citizens' intention to use e-government services.

Regarding the other hypotheses, it was demonstrated that they were supported by the proposed model. This means that the results suggest that higher perceived utility/benefit, better attitude towards use, and greater social influence lead to a higher intention to use e-government. This aligns with the studies of ElKheshin and Saleeb (2017), who state, among others, that citizens' intention to use electronic government services is primarily influenced by their attitude towards using these services due to the voluntary nature of citizens' adoption of e-government services. Additionally, the results contradict the findings of Mensah (2019), who conducted a similar study among university students in the city of Harbin, China, and who claim that subjective norm does not have a positive influence on attitude towards use, meaning that the views, experiences, and opinions of friends, family, and other close circles do not impact individuals' positive attitude towards the use of e-government.

It Is important to emphasize that mobile e-government (or "m-government") has become an optimal option with by to promote the adoption of e-government. According to Al-Hadidi and Rezgui (2010), citizens in emerging economies have more access to smartphones than desktop computers, laptops, and electronic tablets due to their relatively lower cost. This implies that governments need to focus the design of e-government on mobile devices, improving the presentation, ensuring reliability, and guaranteeing the usefulness of the specific functionalities for these platforms.

Furthermore, common and related terms must be taken into account with e-government, for example, the TAM, which served as the basis for the research of Al-Hujran and Shahateet (2010), ElKheshin and Saleeb (2017), Bayaga and Ophoff (2019), and Chukwu et al. (2019). They used TAM constructs and identified new factors for the adoption of ICTs. One of these factors, mentioned even in other studies, is culture. Culture plays an important role in an individual's decision to adopt or reject technology, as concluded by Susanto and Aljoza (2015). Chukwu et al. (2019) agree with that conclusion, stating that usability and data privacy limit the adoption of e-government in Nigeria.

Other authors have employed different models, such as Sabani (2021), who applied the UTAUT model to assess citizen adoption of e-government in Indonesia, confirming

that performance expectancy, effort expectancy, social influence, facilitating conditions, and transparency are critical factors. Meanwhile, Bojang (2021) utilized a proprietary model integrated with the one proposed by DeLone and McLean (2003) to evaluate citizens' intention to adopt e-government initiatives in Gambia, comprehensively considering the forms and means of transacting with the government and the quality of the information received. The study found that raising awareness and providing quality services significantly influence citizens' intention to adopt such services.

These elements, added to others such as a lack of trust in governments and entrenched mentalities (Imran and Gregor 2010), comprise the barriers identified for the implementation of e-government in emerging economies. Many of these factors described in the articles cited coincide with those reported in the Technology and Information Report, 2021, of the United Nations Conference on Trade and Development, in which they also consider issues related to the digital divide as one of the main barriers to the adoption of technologies.

## 6. Implications

This research could assist governments, institutions, decision-makers, policy-makers, and e-government developers in developing countries, particularly in the Latin American context, in identifying the factors that influence the adoption of e-government services among university students.

### 6.1. Implications for Practice

By understanding the unique challenges and opportunities present in Latin America, policy-makers and professionals can develop specific strategies with which to promote the adoption and acceptance of e-government services in the region. Furthermore, the results of this study can serve as a foundation for future research in the field of e-government adoption in Latin America. By addressing the specific context and challenges faced by this region, researchers can continue to explore and expand upon the identified factors and their implications. This can lead to the development of specific models and frameworks for the region that better capture the complexities and nuances of e-government adoption in Latin America.

### 6.2. Theoretical Contributions

In terms of theoretical contributions, this study adds to the existing body of knowledge on e-government adoption by specifically focusing on the Latin American context and the perspective of university students. By highlighting the factors that influence e-government adoption among this demographic group, the study offers valuable insights into the dynamics of technology acceptance and use in Latin America. This contributes to a more comprehensive understanding of the cultural, social, and technological factors that shape e-government adoption in the region.

At the same time, the results obtained underline the relevance of classical models and theories of technology adoption and intention to use, such as TAM or TPB. Thus, a rich theoretical framework is provided which contributes to the production of models of the adoption of digital government technologies in the context of an emerging economy with a strong population that adopts digital technologies and which can help strengthen and increase the use of these technologies.

### 6.3. Managerial Implications

Likewise, the managerial implications for governments and institutions involved in the implementation of e-government services in developing countries, especially in Latin America, emphasize the need to address cultural barriers, usability concerns, data privacy, trust in governments, and entrenched mindsets in order to foster the adoption of e-government services among university students. Implementation stakeholders must consider strategies with which to overcome these barriers, such as promoting education and awareness about the benefits of e-government services, as well as improving security

and data protection. Furthermore, this study underscores the importance of engaging university students as agents of change and advocates for e-government services among their peers and family members.

Thus, the results of this study provide valuable information with which to improve public policies related to the implementation and promotion of e-government in the city of Medellín by supplying the factors that most influence the adoption of this technology from the point of view of university students and establishing the groundwork from which to adjust strategies and encourage greater use of national government platforms. In addition, understanding these factors could drive the areas that require more attention, i.e., through providing training in the use of these tools for citizens.

Finally, an input is provided for the identification of barriers and facilitating aspects with respect to the adoption of e-government in order to promote greater use and transparency in access to information and citizen participation, as well as the reduction of the digital divide in the country by guaranteeing access to ICT for the most vulnerable populations. In this sense, our study could contribute to raising awareness of the importance of digital governance and its benefits for both citizens and the government.

## 7. Conclusions

Hypotheses regarding perceived usefulness and ease-of-use in relation to attitude towards use have not been widely supported in this research context. However, support has been found for hypotheses connecting perceived usefulness with intention to use, attitude towards use with intention to use, perceived ease-of-use with perceived usefulness, intention to use with actual system use, and subjective norm with intention to use. These results suggest that perceived usefulness and ease-of-use are not directly related to attitude towards use, but they do influence intention to use. Additionally, perceived ease-of-use is associated with perceived usefulness, indicating that higher perceived ease of using a system translates into a greater perception of usefulness. Furthermore, intention to use is strongly linked to actual system use, implying that those who have a higher intention to use are more likely to do so effectively. Finally, the subjective norm, or perceived social influence, also affects intention to use. These findings are valuable to understanding the factors that influence the adoption and use of systems and can be leveraged to design effective promotion and persuasion strategies regarding technology adoption.

Notably, according to the results of this study, although most governments, institutions, decision-makers, and developers of e-government measure e-government adoption through the use of indicators, the challenge of stimulating citizen appropriation of this tool is enormous. Therefore, they must carry out specific tasks focused on expanding fixed and mobile internet coverage, exponentially increasing the installation, maintenance, and operation of ICT infrastructure, with an emphasis on rural and remote areas. In addition, digital literacy programs must be designed, implemented, and disseminated to encourage the use of digital tools and eventually facilitate the adoption of e-government. Another task, no less important, is to achieve trust in governments through the success of their actions and the effective dissemination of the good outcomes of government programs and development plans.

A variable that was not taken into account in the research (although it is mentioned) but is extremely detrimental to the adoption of e-government is corruption. Historically, corruption has been prominent in Latin American, and Colombia is no exception. Among the notable points of corruption are patronage and misnamed political quotas. Because of bureaucracy, it is difficult to recognize the government services being offered, where fewer jobs are required due to digital transformation. More common, however, is that instead of having fewer jobs, new ones are created—in some cases, for humans. This is a very large step towards the OECD goal of 100% digital governance.

The lack of adoption of e-government services in Colombia would imply a technological lag compared to other countries that have adopted them. This situation would generate negative indicators for the country's administrators and represent a failure in terms of

government policy. It would also mean a significant loss of resources in terms of planning, time, money, infrastructure, and publicity. It would also be reflected in the increase in slow service and congestion in government offices, as the demand for services grows in parallel with the increase in population.

In this context, university students, due to their educational characteristics and their constant contact with ICT, represent a suitable segment of the population to adopt, use, and promote the use of e-government tools among their relatives and friends. Their familiarity with technology and their ability to disseminate information can contribute significantly to increasing the adoption of these services in society.

Regarding the development and dissemination of electronic government and related mobile and other services, progress has been made, but there is still room for improvement in aspects associated with infrastructure due to persistent connectivity issues in different countries. There is a lack of management in the dissemination of e-government services, limitations in crucial information and services, as well as human factors, including people's limited awareness or lack of knowledge (Furuholt and Sæbø 2018). Additionally, the flexibility and adaptability of the regulations governing these services need to be strengthened, and government activities should be aligned with the needs and interests of citizens (Sararu 2023). Furthermore, there is a need for unification in regulating aspects to increase individuals' trust, build long-term relationships, reduce anxiety or distrust, and increase the intention to use or access digital tools and conduct digital transactions (Gregušová et al. 2022; Zahid and Din 2019).

It is relevant to consider that although this study focuses on developing countries, there are political, social, and economic disparities between the Middle Eastern countries and Western nations. This implies that the findings from the reviewed studies may not necessarily be applicable to the reality and needs of Latin American nations. Additionally, another potential limitation of this research lies in the difficulties encountered when attempting to access the full text of certain articles, which, in turn, hinders access to information that could have allowed for a broader interpretation and understanding of the addressed topic.

For future research, it is suggested that we identify the cultural and digital illiteracy elements that act as barriers to the adoption and appropriation of e-government in emerging Latin American economies. Additionally, it is important to consider specific variables within the Colombian context, such as infrastructure theft, insufficient ICT infrastructure, insecurity, corruption, and the lack of public policies focused on technological adoption. Furthermore, it is recommended that we delve into the adoption and appropriation of innovation and technologies in emerging Latin American economies, as there is a scarcity of articles related to e-government adoption in these countries.

**Author Contributions:** Conceptualization, C.A.M.-R. and O.N.P.-T.; methodology, O.N.P.-T., A.V.-A. and D.M.A.-B.; software, D.M.A.-B.; validation, C.A.M.-R., O.N.P.-T., A.V.-A. and D.M.A.-B.; formal analysis, O.N.P.-T. and D.M.A.-B.; investigation, C.A.M.-R. and O.N.P.-T.; resources, C.A.M.-R. and A.V.-A.; data curation, O.N.P.-T. and D.M.A.-B.; writing—original draft preparation, C.A.M.-R. and O.N.P.-T.; writing—review and editing, C.A.M.-R., O.N.P.-T., A.V.-A. and D.M.A.-B., O.N.P.-T. and D.M.A.-B.; supervision, C.A.M.-R. and O.N.P.-T.; project administration, C.A.M.-R., O.N.P.-T. and A.V.-A.; funding acquisition, C.A.M.-R. and O.N.P.-T. All authors have read and agreed to the published version of the manuscript.

**Funding:** The APC was funded by Universidad Señor de Sipán—USS.

**Informed Consent Statement:** Informed consent was obtained from all subjects involved in the study.

**Data Availability Statement:** The data may be provided free of charge to interested readers by requesting the correspondence author's email.

**Conflicts of Interest:** The authors declare no conflict of interest.

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
