# Peer review of "Factors Influencing the Adoption of E-Government Services: A Study among University Students"

_economies, doi:10.3390/economies11090225_

Round 1
Reviewer 1 Report
Thank you for submitting your manuscript titled "Factors Influencing the Adoption of E-Government Services: A Study among University Students." After careful evaluation, I have identified several areas that require improvement. Please consider the following comments to enhance the quality and clarity of your work:
1. The manuscript lacks a clear and well-defined research problem. The research problem should clearly state the gap in the existing literature or the specific issue that the study aims to address. Without a well-defined research problem, it becomes challenging for readers to understand the significance and relevance of the study.
2. One area of improvement for the manuscript is the organization of the literature review section. It is recommended to separate the literature review section from the introduction section to enhance the clarity and flow of the manuscript. This separation will allow readers to distinguish between the background information provided in the introduction and the review of relevant literature.
3. The manuscript would benefit from a more comprehensive and structured discussion of the hypotheses development. The current presentation of the hypotheses using bullet points (e.g., lines 132 to 198) lacks clarity and makes it difficult for readers to follow the logical progression of the study. For each hypothesis, provide a concise explanation of the underlying theoretical reasoning and empirical evidence, if available. This will help readers understand the logical basis for each hypothesis and its relevance to the research problem. Avoid using bullet points and instead present the hypotheses in paragraph form, using clear and concise language. This will enhance the readability of the manuscript and ensure that the hypotheses are presented in a coherent and logical manner.
4. One major issue with the current manuscript is the lack of clarity and consistency between the proposed research model depicted in Figure 1 and the actual findings of the study. The authors mention that the research model incorporates various factors and theories such as TAM and TPB and other factors. However, the relationships between these factors and the overall model are not adequately discussed or hypothesized in the manuscript. Moreover, compare the proposed research model in Figure 1 with the findings obtained from the data analysis. If there are discrepancies or differences between the proposed model and the empirical results, discuss these inconsistencies and provide possible explanations. It is important to explain the reasons behind any variations or deviations and how they may impact the interpretation and generalizability of the findings.
5. The manuscript has several methodological issues that need to be addressed to ensure the robustness and validity of the study. Here are some specific concerns regarding the methodology:
A. Sampling justification: The manuscript lacks a clear justification for the chosen sampling method and sample size. It is essential to provide a rationale for why the selected sample of 403 university students is representative and appropriate for addressing the research objectives. Discuss any potential limitations or biases associated with the sampling method and how they were mitigated.
B. Data collection process: The manuscript should provide a detailed description of the data collection process. Explain how the survey was administered, who collected the data, and the steps taken to ensure data quality and accuracy. Additionally, address any potential biases or limitations related to the data collection process.
C. Response bias: Consider discussing the potential for response bias in the study. Explain how the authors attempted to mitigate this bias and ensure the accuracy and reliability of the responses obtained from the participants. Discuss any measures taken to encourage honest and unbiased responses.
D. Common Method Bias (CMB): As the study is cross-sectional, there is a possibility of Common Method Bias (CMB) affecting the results. Acknowledge this concern and discuss any steps taken to minimize or control for CMB. This could include the use of counterbalancing techniques, careful design of the survey instrument, or statistical approaches to address CMB.
E. Outliers: Address the presence of outliers in the data and explain how they were managed in the analysis. Describe the criteria used to identify outliers and any decisions made regarding their treatment, such as exclusion or transformation.
F. Estimation method in SEM: Specify the estimation method used in Structural Equation Modeling (SEM) and provide a brief rationale for its selection. Explain how the chosen method is appropriate for analyzing the proposed research model and handling any specific characteristics or assumptions of the data.
6. The discussion section of the manuscript would benefit from additional enrichment by comparing the results with the existing literature and providing a deeper analysis of the findings.
7. Additionally, the manuscript lacks a clear discussion of the implications and theoretical contributions of the study. It is crucial to highlight the significance of the findings and their potential impact on the field of e-government services among university students. By adding a new section contain the following subsections (Implications for practice, Theoretical contributions, Managerial Implications).
8. The conclusion section should include a discussion of the limitations of the study and suggestions for future research. This will provide a balanced view of the research and guide future investigations in addressing the identified gaps.
Extensive editing of English language required
Author Response
July 06, 2023
Dear
Economies – Editorial Team
Kind regards
In accordance with the suggestions of the reviewers in our article “Factors Influencing the Adoption of E-Government Services: A Study among University Students”, the following changes were made, properly marked with red letters in the article:
|
Reviewer |
Comment |
Response |
|
R1 |
1. The manuscript lacks a clear and well-defined research problem. The research problem should clearly state the gap in the existing literature or the specific issue that the study aims to address. Without a well-defined research problem, it becomes challenging for readers to understand the significance and relevance of the study. |
An attempt is made to improve the definition of the research problem, indicating the gap in the existing literature or the specific problem that the study intends to address. |
|
R1 |
2. One area of improvement for the manuscript is the organization of the literature review section. It is recommended to separate the literature review section from the introduction section to enhance the clarity and flow of the manuscript. This separation will allow readers to distinguish between the background information provided in the introduction and the review of relevant literature. |
The introduction and literature review sections were separated |
|
R1 |
3. The manuscript would benefit from a more comprehensive and structured discussion of the hypotheses development. The current presentation of the hypotheses using bullet points (e.g., lines 132 to 198) lacks clarity and makes it difficult for readers to follow the logical progression of the study. For each hypothesis, provide a concise explanation of the underlying theoretical reasoning and empirical evidence, if available. This will help readers understand the logical basis for each hypothesis and its relevance to the research problem. Avoid using bullet points and instead present the hypotheses in paragraph form, using clear and concise language. This will enhance the readability of the manuscript and ensure that the hypotheses are presented in a coherent and logical manner. |
The hypotheses are presented in paragraph form with a concise explanation of the underlying theoretical reasoning and available empirical evidence. |
|
R1 |
4. One major issue with the current manuscript is the lack of clarity and consistency between the proposed research model depicted in Figure 1 and the actual findings of the study. The authors mention that the research model incorporates various factors and theories such as TAM and TPB and other factors. However, the relationships between these factors and the overall model are not adequately discussed or hypothesized in the manuscript. Moreover, compare the proposed research model in Figure 1 with the findings obtained from the data analysis. If there are discrepancies or differences between the proposed model and the empirical results, discuss these inconsistencies and provide possible explanations. It is important to explain the reasons behind any variations or deviations and how they may impact the interpretation and generalizability of the findings. |
The relationships between these factors and the general model are discussed and hypotheses are formulated. In addition, the research model proposed in Figure 1 is compared with the results obtained from the data analysis. Likewise, possible discrepancies or differences between the proposed model and the empirical results are exposed; these inconsistencies are discussed and possible explanations are provided. |
|
R1 |
A. Sampling justification: The manuscript lacks a clear justification for the chosen sampling method and sample size. It is essential to provide a rationale for why the selected sample of 403 university students is representative and appropriate for addressing the research objectives. Discuss any potential limitations or biases associated with the sampling method and how they were mitigated. |
The justification of the chosen sampling method and sample size is extended. It specifies why the selected sample of 403 university students is representative and adequate to address the research objectives and discusses possible limitations or biases associated with the sampling method and how they were mitigated. |
|
R1 |
B. Data collection process: The manuscript should provide a detailed description of the data collection process. Explain how the survey was administered, who collected the data, and the steps taken to ensure data quality and accuracy. Additionally, address any potential biases or limitations related to the data collection process. |
A detailed description of the data collection process is provided. How the survey was administered, who was responsible for collecting the data, and what steps were taken to ensure the quality and accuracy of the data. In addition, possible bias or limitation related to the data collection process is addressed. |
|
R1 |
C. Response bias: Consider discussing the potential for response bias in the study. Explain how the authors attempted to mitigate this bias and ensure the accuracy and reliability of the responses obtained from the participants. Discuss any measures taken to encourage honest and unbiased responses. |
Prior to the application of the questionnaire, a pilot test was carried out with 15 participants to establish clarity in the design of the questionnaire, avoiding possible limitations or biases and determining the relevance of the variables used as the basis for data analysis (Castrillón & Mandakovic, 2010) in the context of e-government adoption. The instrument designed and approved in the pilot test was applied in the presence of one of the authors of this study, to address any concerns that might arise from the students investigated. Emphasis was placed on the importance of responding with total autonomy and tranquility, as it is an anonymous process (Ortiz, et al., 2014). Additionally, to encourage honest and impartial responses from the respondents, simple, non-redundant questions were sought and mixed throughout the instrument to avoid systematic responses. |
|
R1 |
D. Common Method Bias (CMB): As the study is cross-sectional, there is a possibility of Common Method Bias (CMB) affecting the results. Acknowledge this concern and discuss any steps taken to minimize or control for CMB. This could include the use of counterbalancing techniques, careful design of the survey instrument, or statistical approaches to address CMB. |
Harman's one-factor test was used to verify that Common Method Bias did not occur. By reducing the dimensionality to a single factor, an explained variance percentage equal to 30.53 % was reached, which is below the cut-off point of 50% (Podsakoff and Organ, 1986), which is commonly accepted in this type of verification. . Thus, it is concluded that the common method bias is not a problem in our research. |
|
R1 |
E. Outliers: Address the presence of outliers in the data and explain how they were managed in the analysis. Describe the criteria used to identify outliers and any decisions made regarding their treatment, such as exclusion or transformation. |
For the identification of outliers, the data was represented through boxes and whiskers, managing to identify 52 records in common for all the items or questions. This is what was written in the manuscript, based on the comment of the evaluator: |
|
R1 |
F. Estimation method in SEM: Specify the estimation method used in Structural Equation Modeling (SEM) and provide a brief rationale for its selection. Explain how the chosen method is appropriate for analyzing the proposed research model and handling any specific characteristics or assumptions of the data. |
To verify the assumption of normality in the data, we proceeded with the calculation of measures of kurtosis and asymmetry. According to Hu and Bentler (1999), acceptable values to confirm normality are between -2 and 2. The values of the skewness coefficient for each item or question ranged from -0.2 to 0.2, while the values of the kurtosis coefficient ranged from 1.8. and 3.0. For this reason, the "Diagonal Weighted Least Squares" (DWLS) method was used as an estimator to fit the structural equations model with the help of the sem function of the lavaan package of the R software. Studies that support the use of this method for non-normal situations or adjusting ordinal variables are DiStefano and Morgan (2014) and Mindrila (2010), to name just a few. |
|
R1 |
6. The discussion section of the manuscript would benefit from additional enrichment by comparing the results with the existing literature and providing a deeper analysis of the findings. |
Studies were included to compare the results with those of the present study. |
|
R1 |
7. Additionally, the manuscript lacks a clear discussion of the implications and theoretical contributions of the study. It is crucial to highlight the significance of the findings and their potential impact on the field of e-government services among university students. By adding a new section contain the following subsections (Implications for practice, Theoretical contributions, Managerial Implications). |
Added an implications section including theoretical contributions and practical and managerial implications |
|
R1 |
8. The conclusion section should include a discussion of the limitations of the study and suggestions for future research. This will provide a balanced view of the research and guide future investigations in addressing the identified gaps. |
Limitations and suggestions for future research were added in the conclusions |
We look forward to your comments and hope to hear from you soon.
Thank you very much
_
The authors
Reviewer 2 Report
The reviewed article has quite a lot of potential, the topic can be evaluated very positively, but the authors have not sufficiently mastered the guidelines for the authors of this scientific journal.
In the introduction, which is more of a theoretical basis, the reason and goal of the article must be clearly defined, as well as the research questions or hypotheses must be established.
The second chapter is processed in a special way, probably by the author, I would appreciate a better definition of the used scientific research methods as well as the used sources.
The third, focal chapter is, in my opinion, too atomized by dividing it into sub-chapters, and these are completely unnecessarily divided into several-sentence sections. From a content point of view, I would be interested in how the authors envision building e-government and related mobile and other services. In my opinion, this is not a simple issue, but rather a very complicated one. I recommend the authors to also focus on the explanation of this question and they can be inspired by such works of foreign authors as
"Gregušová D, Halásová Z. & Peráček T. (2022). eIDAS Regulation and Its Impact on National Legislation: The Case of the Slovak Republic. Administrative Sciences. 12 (4):187. https://doi.org/10.3390/admsci12040187"
as well as
Săraru, C. S. (2023). Regulation of Public Services in the Administrative Code of Romania: Challenges and Limitations. Access to Justice in Eastern Europe, 18 (1). pp. 69-83. doi:10.33327/AJEE-18-6.1-a000110.
I recommend supplementing the conclusion with answers to established research questions or hypotheses.
Good luck
Author Response
July 06, 2023
Dear
Economies – Editorial Team
Kind regards
In accordance with the suggestions of the reviewers in our article “Factors Influencing the Adoption of E-Government Services: A Study among University Students”, the following changes were made, properly marked with red letters in the article:
|
Reviewer |
Comment |
Response |
|
R2 |
In the introduction, which is more of a theoretical basis, the reason and goal of the article must be clearly defined, as well as the research questions or hypotheses must be established. |
The reason and objective of the article are defined in the Introduction of the document and the research hypotheses are established. |
|
R2 |
The second chapter is processed in a special way, probably by the author, I would appreciate a better definition of the used scientific research methods as well as the used sources. |
It seeks to improve the definition and description of the scientific research methods used as well as the sources used. |
|
R2 |
The third, focal chapter is, in my opinion, too atomized by dividing it into sub-chapters, and these are completely unnecessarily divided into several-sentence sections. From a content point of view, I would be interested in how the authors envision building e-government and related mobile and other services. In my opinion, this is not a simple issue, but rather a very complicated one. I recommend the authors to also focus on the explanation of this question and they can be inspired by such works of foreign authors as "Gregušová D, Halásová Z. & Peráček T. (2022). eIDAS Regulation and Its Impact on National Legislation: The Case of the Slovak Republic. Administrative Sciences. 12 (4):187. https://doi.org/10.3390/admsci12040187" as well as Săraru, C. S. (2023). Regulation of Public Services in the Administrative Code of Romania: Challenges and Limitations. Access to Justice in Eastern Europe, 18 (1). pp. 69-83. doi:10.33327/AJEE-18-6.1-a000110. |
The division by subchapters is eliminated and improvements or supplements are included in the writing for a better understanding of the text. In relation to the content, a paragraph is included with information |
|
R2 |
I recommend supplementing the conclusion with answers to established research questions or hypotheses. |
The conclusions were complemented with a paragraph where an answer is given to the hypotheses initially raised in the study. |
We look forward to your comments and hope to hear from you soon.
Thank you very much
_
The authors
Round 2
Reviewer 1 Report
Dear Authors,
Thank you for revising your manuscript. After carefully examining the revised version, I am still finding several issues that need to be addressed to ensure a clear presentation and rigorous analysis of your research. Here are the main points that need to be improved:
1. Your manuscript still lacks a clear and well-defined research problem. The research problem should be clearly stated to articulate the gap in the existing literature or the specific issue that your study aims to address. At the moment, it remains difficult for readers to understand the importance and relevance of your study without a well-formulated problem statement. I recommend you revisit this aspect of the paper and define a clear problem that your research aims to address.
2. There is still a lack of structure and detail in your hypotheses development section. As previously mentioned, your current presentation of the hypotheses lacks clarity and makes it difficult for readers to follow your study's logical progression. For each hypothesis, a detailed explanation of the theoretical reasoning and empirical evidence supporting it, if available, should be provided. Please refrain from using bullet points; instead, present the hypotheses in paragraph form, using clear and concise language. This will significantly enhance the readability and coherence of your manuscript.
3. Concerning Figure 1, it still needs to be redrawn to clearly illustrate the constructs and the associated hypotheses numbers. This would aid in readers' understanding and would map out the logical flow of your argument more effectively.
4. The structure of the Methodology section, particularly the justification for using the analysis approach and the chosen instrument, still needs improvement. Moreover, the subsection titled "Limitations associated with the sampling method" is written more in a textbook style than an academic paper style. Please revise this section to adopt a more scientific tone and provide a clear justification for your methodological choices.
5. The newly added Implications section appears to lack depth. It would be useful to elaborate more on the practical implications of your findings for practitioners and policymakers in the field of e-government services. Moreover, the section could further discuss how your study contributes to the theoretical understanding of e-government adoption.
6. Lastly, the overall writing style of your manuscript could be improved to make it more precise and engaging. I noticed some instances of awkward sentence structure and overly complex language. These issues impede readability and comprehension. It is recommended to go through your manuscript thoroughly and edit it for clarity, coherence, and conciseness.
Addressing these issues will greatly improve the quality and readability of your manuscript, enhancing its potential contribution to the field. I look forward to your revised manuscript.
Extensive editing of the English language required
Author Response
August 01, 2023
Dear
Economies – Editorial Team
Kind regards
In accordance with the suggestions of the reviewers in our article “Factors Influencing the Adoption of E-Government Services: A Study among University Students”, the following changes were made, properly marked with red letters in the article:
|
Comment |
Response |
|
1. Your manuscript still lacks a clear and well-defined research problem. The research problem should be clearly stated to articulate the gap in the existing literature or the specific issue that your study aims to address. At the moment, it remains difficult for readers to understand the importance and relevance of your study without a well-formulated problem statement. I recommend you revisit this aspect of the paper and define a clear problem that your research aims to address. |
The research problem has been clearly stated, articulating the gap in existing literature and the specific issue addressed by the study. |
|
2. There is still a lack of structure and detail in your hypotheses development section. As previously mentioned, your current presentation of the hypotheses lacks clarity and makes it difficult for readers to follow your study's logical progression. For each hypothesis, a detailed explanation of the theoretical reasoning and empirical evidence supporting it, if available, should be provided. Please refrain from using bullet points; instead, present the hypotheses in paragraph form, using clear and concise language. This will significantly enhance the readability and coherence of your manuscript. |
The hypotheses development section has been revised to provide more structure and detail. Each hypothesis now includes a detailed explanation of theoretical reasoning and empirical evidence, presented in paragraph form for better coherence. |
|
3. Concerning Figure 1, it still needs to be redrawn to clearly illustrate the constructs and the associated hypotheses numbers. This would aid in readers' understanding and would map out the logical flow of your argument more effectively. |
Figure 1 has been redrawn to clearly illustrate the constructs and associated hypotheses numbers, aiding readers' understanding of the logical flow. |
|
4. The structure of the Methodology section, particularly the justification for using the analysis approach and the chosen instrument, still needs improvement. Moreover, the subsection titled "Limitations associated with the sampling method" is written more in a textbook style than an academic paper style. Please revise this section to adopt a more scientific tone and provide a clear justification for your methodological choices. |
The Methodology section has been improved, providing a more scientific tone and clear justification for the analysis approach and chosen instrument. The subsection on limitations has also been revised accordingly. |
|
5. The newly added Implications section appears to lack depth. It would be useful to elaborate more on the practical implications of your findings for practitioners and policymakers in the field of e-government services. Moreover, the section could further discuss how your study contributes to the theoretical understanding of e-government adoption. |
The Implications section has been expanded to elaborate on the practical implications for practitioners and policymakers in e-government services. Additionally, the study's contribution to theoretical understanding is discussed further. |
|
6. Lastly, the overall writing style of your manuscript could be improved to make it more precise and engaging. I noticed some instances of awkward sentence structure and overly complex language. These issues impede readability and comprehension. It is recommended to go through your manuscript thoroughly and edit it for clarity, coherence, and conciseness. |
The overall writing style has been refined for precision and engagement. Awkward sentence structures and overly complex language have been addressed to enhance readability and comprehension. |
We look forward to your comments and hope to hear from you soon.
Thank you very much.
_
The authors
Reviewer 2 Report
I am glad that the authors incorporated all comments and therefore I recommend the article for publication in this form.
Author Response

(The authors gave the same response as above.)

Round 3
Reviewer 1 Report
I commend the authors for the significant improvements made in the paper compared to the previous versions. However, there are still some crucial areas that require attention. Addressing the following suggestions will undoubtedly enhance the overall quality and contribution of your research.
1. The introduction section has been strengthened, but there are several claims made without sufficient supporting evidence from recent references. Please ensure that all statements made between lines 57 and 94, lines 104 and 107, and lines 114 and 136 are backed by relevant and recent scholarly sources. Providing empirical evidence will increase the credibility of your paper.
2. In section 2.1.3, discussing models of innovation adoption using bullet points (lines 188 to 237) may not be the most suitable approach. I recommend presenting these models and theories in a narrative form to ensure a smooth flow of ideas. Additionally, support each mentioned model or theory with recent studies that have employed them in their research. This will strengthen the theoretical foundation of your study.
3. Regarding section 2.2, please rename it to "2.2 Hypotheses Development" to better reflect its content. Under this section, present all seven hypotheses in separate subsections. For example, you can use the format of "H1: Construct1 (IV) has a positive and significant impact on Construct2 (DV)" and discuss each hypothesis in detail. Moreover, consider redrawing Figure 1, labeling the full names of the constructs, and adding hypothesis numbers to Table 8 (Estimates and Hypothesis Testing) in section 4.3. This will provide clarity and a better understanding of your research framework.
4. Consider merging the content of Table 4 (Model Reliability Test) with Table 5 (Factor Loadings for the Theoretical Model). This will make the presentation more concise and avoid redundancy in the paper.
Moderate editing of the English language required
Author Response
August 06, 2023
Dear
Economies – Editorial Team
Kind regards
In accordance with the suggestions of the reviewers in our article “Factors Influencing the Adoption of E-Government Services: A Study among University Students”, the following changes were made, properly marked with blue letters in the article:
|
Comment |
Response |
|
1. The introduction section has been strengthened, but there are several claims made without sufficient supporting evidence from recent references. Please ensure that all statements made between lines 57 and 94, lines 104 and 107, and lines 114 and 136 are backed by relevant and recent scholarly sources. Providing empirical evidence will increase the credibility of your paper. |
Quotations were added in the indicated lines to support the text |
|
2. In section 2.1.3, discussing models of innovation adoption using bullet points (lines 188 to 237) may not be the most suitable approach. I recommend presenting these models and theories in a narrative form to ensure a smooth flow of ideas. Additionally, support each mentioned model or theory with recent studies that have employed them in their research. This will strengthen the theoretical foundation of your study. |
The bullets were omitted, making the writing fluid and adding recent studies to support the validity of the theories. |
|
3. Regarding section 2.2, please rename it to "2.2 Hypotheses Development" to better reflect its content. Under this section, present all seven hypotheses in separate subsections. For example, you can use the format of "H1: Construct1 (IV) has a positive and significant impact on Construct2 (DV)" and discuss each hypothesis in detail. Moreover, consider redrawing Figure 1, labeling the full names of the constructs, and adding hypothesis numbers to Table 8 (Estimates and Hypothesis Testing) in section 4.3. This will provide clarity and a better understanding of your research framework. |
Changed the name 2.2 Hypothesis Development" to better reflect its content. |
|
4. Consider merging the content of Table 4 (Model Reliability Test) with Table 5 (Factor Loadings for the Theoretical Model). This will make the presentation more concise and avoid redundancy in the paper. |
The two tables were merged |
We look forward to your comments and hope to hear from you soon.
Thank you very much.
_
The authors